# Chrysophanol Mitigates T Cell Activation by Regulating the Expression of CD40 Ligand in Activated T Cells

**DOI:** 10.3390/ijms21176122

**Published:** 2020-08-25

**Authors:** Hyun-Su Lee, Gil-Saeng Jeong

**Affiliations:** College of Pharmacy, Keimyung University, Daegu 42601, Korea; hyunsu.lee@kmu.ac.kr

**Keywords:** T cells, IL-2, CD40L, immunological synapse, MAPK

## Abstract

Since T lymphocytes act as mediators between innate and acquired immunity, playing a crucial role in chronic inflammation, regulation of T cell activation to suitable levels is important. Chrysophanol, a member of the anthraquinone family, is known to possess several bioactivities, including anti-microbial, anti-cancer, and hepatoprotective activities, however, little information is available on the inhibitory effects of chrysophanol on T cell activation. To elucidate whether chrysophanol regulates the activity of T cells, IL-2 expression in activated Jurkat T cells pretreated with chrysophanol was assessed. We showed that chrysophanol is not cytotoxic to Jurkat T cells under culture conditions using RPMI (Rosewell Park Memorial Institute) medium. Pretreatment with chrysophanol inhibited IL-2 production in T cells stimulated by CD3/28 antibodies or SEE-loaded Raji B cells. We also demonstrated that chrysophanol suppressed the expression of the CD40 ligand (CD40L) in activated T cells, and uncontrolled conjugation between B cells by pretreatment with chrysophanol reduced T cell activation. Besides, treatment with chrysophanol of Jurkat T cells blocked the NFκB signaling pathway, resulting in the abrogation of MAPK (mitogen-activated protein kinase) in activated T cells. These results provide novel insights into the suppressive effect of chrysophanol on T cell activation through the regulation of CD40L expression in T cell receptor-mediated stimulation conditions.

## 1. Introduction

T cells play a critical role in inflammatory responses, wherein several immune cells are involved, and orchestrate the regulation of immune responses. Pathogenesis of various autoimmune disorders, including atopic dermatitis, immune bowel diseases, asthma, systemic lupus erythematosus, and psoriasis is implicated in the activated T cells [1]. Excessively activated T cells primed with self-antigens can lead to severe autoimmune responses. Several therapeutics for autoimmune disorders have been developed to control undesirable T cell activity in immune responses. Consequently, regulation of excessively activated T cells is pivotal for maintaining immunological homeostasis.

CD40 ligand (CD40L) is a surface protein that is expressed on resting and activated T cells and shows induced patterns in activated T cells [2]. CD40L is known to be required for generating primary cytotoxic T cells in viral infections [3]. Deficiency or mutation of CD40L causes severe immunodeficiency and X-linked Hyper-IgM syndrome [4]. It binds to CD40 expressed on antigen presenting cells (APCs) in the early phase of immunological synapse (IS) and acts to transmit co-stimulatory signals to both T cells and APCs, which in turn help B cells proliferate and differentiate into plasma cells [5]. It is well known that sufficient signals between T cells and APCs promotes mature IS in the late phase of T cell activation. Although primed T cells from APCs are physically separated from T-APC conjugates and differentiated into effector T cells, only a few investigators have explored whether CD40L plays a critical role in the separation of T cells in the late phase of T cell activation, subsequently leading to the regulation of T cell activation.

Chrysophanol, a phytochemical, also known as chrysophanic acid, is a member of the anthraquinone family that has been found in several plants, including *Rheum palmatum* [6], *Cassia fistula* [7], *Aloe excels* [8], and *Cluytia hirsute* [9]. It possesses biological activities such as antitumor [10] and anti-diabetic activities [11], as well as preventive effects on memory and learning functions in Alzheimer’s disease mouse model [12]. In particular, anti-inflammatory effects of chrysophanol on dextran sulfate sodium (DSS)-induced colitis and lipopolysaccharide (LPS)-induced inflammation has been demonstrated to effectively suppress overall clinical concentrations of moieties, including those of interleukin-6 (IL-6), tumor necrosis factor-α (TNF-α), and cyclooxygenase-2 (COX-2) through the regulation of NFκB pathway [13]. Despite its protective activity against LPS-induced inflammation, little is known whether chrysophanol has a suppressive effect on T cell activation. Here, we explored whether chrysophanol controls T cell activation mediated by T cell receptors and its underlying mechanism of action through the regulation of CD40L expression and function.

## 2. Results

### 2.1. Chrysophanol Is Not Cytotoxic to Jurkat T Cells under Culture Conditions Using RPMI Medium

Chrysophanol (Figure 1), a member of the anthraquinone family, has been shown to possess anti-cancer activity, since it regulates proliferation and brings about apoptosis of cancerous cells [10]. In particular, chrysophanol has been reported to cause cytotoxicity and pro-apoptotic activities in Jurkat T cells cultured in DMEM (Dulbecco’s Modified Eagle Medium) medium [14]. By contrast, several studies have reported that chrysophanol does not exhibit cytotoxic effects and protect cells from critical damages [15,16,17]. To clarify whether treatment with chrysophanol exhibits cytotoxicity on Jurkat T cells cultured using different conditions as previously reported, we performed an MTT (3-(4,5-dimethylthiazol-2-yl)-2,5-diphenyl tetrazolium bromide) assay by comparing different media (RPMI (Rosewell Park Memorial Institute) versus DMEM) and different densities of cells (2 × 10^4^/mL to 1 × 10^5^/mL). Figure 2A revealed that 40 μM chrysophanol did not exert cytotoxic effects on Jurkat T cells cultured in RPMI and DMEM at a density of 1 × 10^5^/mL but displayed mild cytotoxicity to Jurkat T cells cultured only in DMEM at a density of 5 × 10^4^/mL or 2 × 10^4^/mL. To obtain growth rate of Jurkat T cells in the presence of 40 μM chrysophanol, we counted the number of Jurkat T cells cultured in these two media every 24 h. As shown in Figure 2B, Jurkat T cells cultured in DMEM showed a significant decrease in growth rate compared to Jurkat T cells cultured in RPMI. To confirm whether the population of apoptotic cells induced by treatment with chrysophanol is dependent on culture media and cell number, AnnexinV/PI (Propidium Iodide) apoptosis assay was performed. Jurkat T cells cultured in DMEM exhibited an increased apoptotic population compared to Jurkat T cells cultured in RPMI, however, treatment with chrysophanol has no pro-apoptotic at the density of 1 × 10^5^/mL. These results suggest that chrysophanol does not cause cell death and apoptosis in Jurkat T cells cultured in RPMI medium.

### 2.2. Chrysophanol Inhibits IL-2 Production from Activated T Cells

To elucidate whether chrysophanol suppresses the activity of T cells in conditions of stimulation mediated by T cell receptors (TCRs), the mRNA level of *il2* and interleukin (IL-2), a marker of early T cell activation produced from activated T cells were determined. Pretreatment of Jurkat T cells activated by immobilized CD3 and CD28 antibodies with chrysophanol reduced the expression of *il2* gene in a dose-dependent manner (Figure 3A). Moreover, the amount of IL-2 produced from activated T cells was also reduced in the presence of chrysophanol in a time- or dose-dependent manner (Figure 3B). B cells, one of the antigen-presenting cells, provide priming signals that initiate T cell activation by forming immunological synapses [18]. To confirm whether chrysophanol also has a suppressive effect on T cell activation by interacting with B cells, Raji B cells pulsed with staphylococcal enterotoxin E (SEE), a superantigen, were used to activate Jurkat T cells. Figure 3C shows that pre-treatment with chrysophanol hinders the mRNA level of *il2* in Jurkat T cells conjugated with SEE-loaded Raji B cells in a dose-dependent manner. In Jurkat T cells conjugated with Raji B cells, the IL-2 production was gradually downregulated depending on the concentration and incubation time of pretreated chrysophanol (Figure 3D). To address whether the inhibited T cell activity by pre-treatment with 40 μM chrysophanol is reversed or recovered as the level of un-treated cells, we tested the removal effect of chrysophanol after pre-treatment with chrysophanol by comparing mRNA level of *il2* from activated T cells pre-treated with chrysophanol for 30 min or 1 h only and activated T cells post-washed with fresh media after pre-treated with chrysophanol for 30 min or 1 h. As shown in Figure 3E, pre-treatment with chrysophanol for 30 min and 1 h of T cells exerted significant decrease of mRNA level of *il2* but removal of chrysphanol after pre-treatment did not show any attenuation in mRNA level of *il2*. Interestingly, decrease of mRNA level of *il2* is dependent on the time of pre-treatment with chrysophanol, meaning that pre-treatment with chrysophanol for 1 h exhibited more inhibitory effect on T cell activation than pre-treatment for 30 min. These data imply that T cell activity is irreversibly restrained by pretreatment with chrysophanol in Jurkat T cells stimulated by antibodies against CD3/CD28 and SEE-pulsed Raji B cells.

### 2.3. Chrysophanol Suppresses the Expression of CD40L in Activated T Cells

CD40L is an inducible surface molecule in activated conditions and plays a crucial role as co-stimulatory molecule in the initiation as well as maintenance of T cell activation [2,5]. Since chrysophanol was observed to have an inhibitory effect on the activity of T cells conjugated with B cells, we investigated whether the expression of CD40L was affected by pretreatment with chrysophanol on activated T cells. Reduced mRNA level of *cd40l* was observed in activated T cells pretreated with 40 μM chrysophanol depending on the incubation time (Figure 4A). To measure the protein level of CD40L in activated T cells in the presence of chrysophanol, Western blot and flow cytometry analyses were performed. Western blotting results exhibited that pretreatment with chrysophanol downregulated CD40L expression in whole lysates of T cells in a dose-dependent manner (Figure 4B). Besides, the suppressive effect of chrysophanol on induction of CD40L expression on the surface of T cells was assessed by flow cytometry depending on stimulation time (Figure 4C). The highest expression of CD40L was observed in T cells stimulated for 24 h. These data suggest that the expression of CD40L on activated T cells is gradually increased by stimulation with CD3 and CD28 antibodies, but pretreatment with chrysophanol inhibits CD40L expression in the whole lysate and on surfaces of T cells.

### 2.4. Abnormal Expression of CD40L by Chrysophanol Reduces T Cell Activation through Uncontrolled Conjugation between T and B Cells

Since expressed CD40L on naïve T cells provides co-stimulatory signaling for the initiation of T cell activation, we investigated whether chrysophanol affects the formation of conjugates of T cells with B cells in the early phase (30 min) and late phase (24 h) of T cell activation. The highest conjugation between T and B cells was observed within 30 min but no significant alteration due to pretreatment with chrysophanol was observed in cells incubated for 30 min (Figure 5A,B). Conjugation was also detected in cells incubated for 24 h but was downregulated compared to cells incubated for 30 min. However, enhanced conjugation was observed in cells incubated for 24 h in the presence of chrysophanol than in control cells. To examine whether different induction of CD40L on the surface of T cells by pretreatment with chrysophanol leads to the increment of conjugation in cells pretreated with chrysophanol, CD40L antibodies were added at the initiation of incubation to neutralize CD40L. Interestingly, conjugation was significantly enhanced in cells incubated with CD40L antibodies and reached comparable levels as that in cells pretreated with chrysophanol (Figure 5A,B). To understand the effect of CD40L expression on T cell activation, the production of IL-2 was assessed by conjugating T cells with B cells and incubating with CD40L neutralizing antibodies. The amount of IL-2 produced was suppressed in T-B conjugates treated with CD40L neutralizing antibodies, and this inhibitory effect was similar to that of pretreatment with chrysophanol (Figure 5C). Microscopic images were obtained using an IncuCyte imaging system to confirm whether restricted expression of CD40L by chrysophanol affects the conjugation between T and B cells (Figure 5D,E). These data imply that the induced CD40L on T cell surface plays a pivotal role in the regulation of immunological synapses in terminal phase. However, pretreatment of T cells with chrysophanol fails to regulate conjugation of T and B cells by downregulating CD40L expression in T cells, resulting in the reduction of T cell activation.

### 2.5. Treatment with Chrysophanol Blocks NFκB Signaling Pathway in Activated T Cells

Among the several signaling pathways associated with IL-2 production in activated T cells, the NFκB pathway has been considered as one of the central modulators. To examine whether the inhibitory effect of chrysophanol on T cell activation is involved in the NFκB pathway, p65 translocation was determined in activated T cells pretreated with chrysophanol. Western blot results revealed that p65 in the cytosol was translocated by stimulation with CD3/CD28 antibodies, but pretreatment with chrysophanol hindered this manifestation (Figure 6A). Degradation of IκBα by T cell activation was also suppressed due to pretreatment with chrysophanol. Besides, pretreatment with chrysophanol downregulated the phosphorylation level of IκBα in activated T cells. These data suggest that chrysophanol blocks the NFκB pathway in activated T cells.

### 2.6. Pre-Treatment with Chrysophanol Abrogates MAPK Signaling Pathway in Activated T Cells

To investigate whether chrysophanol suppresses the MAPK (mitogen activated protein kinase) signaling pathway in activated T cells, phosphorylation levels of ERK (extracellular signal-regulated kinases), p38, JNK (c-Jun *N*-terminal kinases), and c-Jun were detected in time- and dose- dependent manner. Phosphorylation levels of ERK, p38, JNK, and c-Jun were upregulated by CD3/CD28 antibodies within 30 min, but pretreatment with chrysophanol partially abrogated these levels in activated T cells (Figure 7A). A dose-dependent experiment also demonstrated that pretreatment with chrysophanol exhibits impeded phosphorylation of ERK, p38, JNK and c-Jun in activated T cells (Figure 7B). These data imply that the MAPK pathway is involved in suppressive effects of chrysophanol on T cell activation by CD3/CD28 antibodies.

## 3. Discussion

In the present study, we explored whether chrysophanol, a member of the anthraquinone family, has an inhibitory effect on TCR-mediated activation of T cells by using Jurkat T cells. Pretreatment with chrysophanol effectively regulated IL-2 production in activated T cells without displaying any cytotoxic effects under culture condition using RPMI medium. In the late phase of T cell activation, expression of CD40L was downregulated by pretreatment of activated T cells with chrysophanol. This treatment inhibited the separation of T cells from IS in the late phase of conjugation of T cells with B cells, thereby suppressing T cell activation. Furthermore, results of Western blotting analysis indicated that the inhibitory effect of chrysophanol on T cell activation was highly mediated by a decrease in NFκB and MAPK pathways.

It is well known that cell viability assays including MTT and CCK-8 (cell counting kit-8) assays are highly dependent on the number of seeded cells and culture conditions including the types of media used and presence or absence of other agents such as fetal bovine serum (FBS). RPMI is known to have a lower concentration of calcium than DMEM. This culture condition is more suitable for suspension cell cultures such as that of Jurkat T cells or Raji B cells. Nevertheless, a previous study has reported that chrysophanol exhibits cytotoxic effects and promotes an apoptotic pathway in Jurkat T cells cultured in DMEM medium [14]. To check this discrepancy, we compared our culture conditions to those described in previous reports. We found that Yin J and colleagues cultured Jurkat T cells in DMEM medium, performed cytotoxicity assays with different densities of cells, and obtained Jurkat cells from different companies. We cultured Jurkat T cells in RPMI medium according to the manufacturer’s instructions and performed experiments at a density of 1 × 10^4^/200 μL/well. Under these conditions, we assessed whether treatment with 40 μM chrysophanol for 24 h has cytotoxic effects in Jurkat T cells and performed several experiments to demonstrate the regulatory effect of chrysophanol on T cell activity. The data thus obtained suggest that chrysophanol not only has anticancer activities in various cells, but also shows anti-inflammatory effects on several cell lines, including Jurkat T cells used in the present study.

IS shown to be a distinct structure wherein several molecules play a critical role in T cell and APC activation [19]. A central supramolecular activation cluster (cSMAC) has been identified to be the most important part of IS in which pivotal molecules, including TCR, Lck, ZAP70, PKCθ, and CD28, accumulate for recognition of peptide-MHCII complex. CD40L is shown to be localized in the cSMAC of resting T cells and ligates with CD40 expressed on APCs [19]. A recent study demonstrated that CD40L-CD40 conjugation in cSMAC is dependent on the presence of TCR signaling, binding of intracellular adhesion molecule 1 (ICAM-1) to integrin molecules expressed on T cells, and rearrangement of the cytoskeleton in T cells [19]. Although evidence explaining the mechanism of IS termination has been demonstrated, depletion of TCR recycling process induces separation of T cells from conjugation [20] and blockade of CD40L-controlled termination of immune synapse formation (Figure 5). This result assumes that conjugation between T cells and APCs is tightly regulated so as not to be activated by the induction of CD40L in the late phase of T cell activation. Furthermore, pretreatment with chrysophanol suppressed the expression of CD40L in activated T cells and abolished termination of contact between T cells and APCs. Based on these results, the present study implies that chrysophanol has a potential to be developed as an immunosuppressive agent by targeting CD40L.

To activate resting T cells in vitro, a direct stimulation method mediated by TCR is widely used. A mixture of superantigen-loaded B cells and T cells is commonly used to stimulate T cells in an in vitro experiment [21]. It has been developed to mimic the actual situation where TCR engages peptide-MHC complex from B cells in the IS. In this procedure, in vitro immune synapse formation can be assessed using Jurkat T cells and SEE-loaded Raji B cells [22]. The other method to provide T cell-activating signals is to use monoclonal antibodies against CD3 and CD28 constituting TCR. Immobilized CD3 antibodies are potent stimulators that target CD3 without employing the peptide-MHC complex [23]. To supply co-stimulatory signals with CD3 stimulation, soluble CD28 antibodies are orchestrated. The current study revealed that Jurkat T cells were activated by SEE-loaded Raji B cells and antibodies against CD3 and CD28. A tremendous increment of IL-2 levels and phosphorylated MAPK signaling molecules from activated Jurkat T cells demonstrated that the in vitro stimulatory method mediated by TCR triggers phosphorylation of ERK, p38, JNK, and c-Jun, in turn promoting the production of IL-2. To investigate whether chrysophanol has a suppressive effect on T cell activation in vivo, further studies should investigate if oral administration of chrysophanol attenuates manifestations of T cell-mediated diseases such as atopic dermatitis, immune bowel disease, or asthma using an animal model.

## 4. Materials and Methods

### 4.1. Cells

Jurkat T cells and Raji B cells were purchased from the Korean Cell Line Bank (Seoul, Republic of Korea). Cells were grown in RPMI medium (Welgene, Gyeongsan, Republic of Korea) supplemented with 10% fetal bovine serum (FBS), streptomycin (100 μg/mL), penicillin G (100 units/mL) and l-glutamine (2 mM). Cells were grown at 37 °C in a humidified incubator containing 5% CO_2_ and 95% air.

### 4.2. Reagents and Antibodies

MTT (1-(4,5-Dimethylthiazol-2-yl)-3,5-diphenylformazan) powder, CMFDA (5-chloromethylfluorescein diacetate) and CMRA (5-(((4-chloromethyl)benzoyl)amino) tetramethylrhodamine) cell staining dyes, was obtained from Sigma Chemical Co. (St. Louis, MO, USA). Anti-CD3 antibodies and anti-CD28 antibodies were purchased from Bioxcell (West Lebanon, NH, USA). Human IL-2 DuoSet^®^ ELISA kit was obtained from R&D Systems (Minneapolis, MN, USA). Staphylococcal enterotoxin E (SEE) was purchased from Toxin Technology (Sarasota, FL, USA). NE-PER Nuclear and Cytoplasmic Extraction Reagents Kit and ECL Western blotting detection reagents were obtained from Thermo Fisher Scientific (Waltham, MA, USA). Anti-CD40L antibodies conjugated with APC was purchased from eBiosciences (San Diego, CA, USA). Anti-CD40L neutralizing antibodies were obtained from InvivoGen (San Diego, CA, USA). Anti-CD40L for western blot and anti-β-actin antibodies were purchased from Santa Cruz Biotechnology (Dallas, TX, USA). Anti-p65, anti-PARP, anti-IκBα, anti-phosphorylated IκBα (S32), anti-phosphorylated ERK (T202/Y204), anti-ERK, anti-phosphorylated p38 (T180/Y182), anti-p38, anti-phosphorylated JNK (T183/Y185), anti-JNK, anti-phosphorylated c-Jun (S73) and anti-c-Jun antibodies were obtained from Cell Signaling Technology (Danvers, MA, USA).

### 4.3. Isolation of Chrysophanol from Rumex crispus L

Chysophanol used in the current study was isolated from dried leaves of *R. crispus* L. as previously reported in our group [24]. Briefly, dried leaves of *R. crispus* L. (1 kg) were extracted 3 times with 70% ethanol for 24 h under room temperature. The EtOH extract (230 g) was suspended in H_2_O (1 L) and then partitioned with n-hexane (1 L) and EtOAc (1 L). Among the n-hexene (4.5 g) and EtOAc fraction (27.1 g), chrysophanol was purified by repeated open column chromatography on silica gel (6.5 × 60 cm; 70–230 mesh). The isolated chrysophanol was analyzed by ^1^H and ^13^C-NMR (JEOL JNM-ECA 500) NMR spectroscopic, and spectral data were identified as chrysophanol by comparison with the published literature [25].

### 4.4. MTT Assay

Two different media (RPMI or DMEM) cultured-Jurkat T cells (1 × 10^5^–2 × 10^4^/mL) were seeded and incubated with the indicated concentration of chrysophanol (0–40 μM) for 24 h. The supernatants were discarded and cells were added with 500 μg/mL of MTT for 1.5 h. Supernatants were removed and 150 μL of DMSO were added for dissolving formazan crystals. Absorbance was obtained at 540 nm and cell viability was calculated with absorbance of control (% of control).

### 4.5. Assessment of Growth Rate by Cell Counting

Two different media (RPMI or DMEM) cultured-Jurkat T cells (1 × 10–2 × 10^4^/mL) were seeded and incubated with 40 μM chrysophanol for the indicated time. The number of cells was counted every 24 h and was presented in line graph.

### 4.6. AnnexinV/PI Apoptosis Assay by Flow Cytometry

Two different media (RPMI or DMEM) cultured-Jurkat T cells (1 × 10^5^–2 × 10^4^/mL) were seeded and incubated with the indicated concentration of chrysophanol (0–40 μM) for 24 h. After incubation, cells were stained with AnnexinV and PI and apoptotic cells were acquired by flow cytometry. The population of apoptotic cells were presented in dot plots.

### 4.7. T Cell Stimulation

Jurkat T cells (5 × 10^5^/mL) were pre-treated with chrysophanol at the indicated concentration (0–40 μM) or 40 μM for 1 h at 37 °C. Then cells were stimulated with plates coated with anti-CD3 (20 μg/mL) and soluble anti-CD28 (7 μg/mL) for the indicated time (6–24 h). For superantigen stimulation, Raji B cells were prepared by pulsing 1 μg/mL SEE for 30 min at 37 °C. Then Jurkat T cells were co-cultured with SEE-pulsed Raji B cells for the indicated time (6–24 h).

### 4.8. Determination of mRNA Level by Conventional and Quantitative PCR

Total RNA was isolated using TRIZOL reagent (Thermo Fisher Scientific, Waltham, MA, USA) and reverse transcription of the RNA to cDNA was performed using RT PreMix (Enzynomics, Daejeon, Korea). Primers used for each gene were as follows (forward and reverse primers, respectively): human *il2*, 5′-CAC GTC TTG CAC TTG TCA C-3′ and 5′-CCT TCT TGG GCA TGT AAA ACT-3′; human *cd40l*, 5′-GCC AGT TTG AAG GCT TTG TG-3′ and 5′-ACT TAT GAC ATG TGC CGC AA-3′, human *gapdh*, 5′-CGG AGT CAA CGG ATT TGG TCG TAT-3′ and 5′-AGC CTT CTC CAT GGT GGT GAA GAC-3′. The condition of conventional PCR was as follows: 28 cycles of denaturation at 94 °C for 30 s, annealing at 60 °C for 20 s, and extension at 72 °C for 40 s; followed by denaturation at 72 °C for 7 min. For quantitative PCR analysis, PCR amplification was performed in a DNA Engine Opticon 1 continuous fluorescence detection system (MJ Research, Waltham, MA, USA) by using SYBR Premix Ex Taq (Takara, Japan). The total reaction volume was 10 μL containing 1 μL of cDNA/control and gene specific primers. Each PCR reaction was performed using the following conditions: 95 °C for 30 s, 60 °C for 30 s, and plate read (detection of fluorescent product) for 38 cycles followed by 7 min of extension at 72 °C. Melting curve analysis was performed to characterize the dsDNA product by slowly raising the temperature (0.2 °C/s) from 60 to 95 °C with fluorescence data collected at 0.2 °C intervals. mRNA levels of *il2* and *cd40l* were normalized with mRNA levels of *gapdh* and were presented as % of controls. The fold change in gene expression was calculated using Equation (1):(1)fold change=2−ΔΔCT
where ΔΔ*CT* = (CTtarget − CTgapdh) at time *x* − (CTtarget − CTgapdh) at time 0. Here, time *x* represents any time point and time 0 represents the 1× expression of the target gene in the control cells normalized to *gapdh*.

### 4.9. IL-2 Measurement by ELISA

Jurkat T cells (1 × 10^5^/mL) were stimulated with immobilized anti-CD3 antibodies (20 μg/mL) and soluble anti-CD28 (7 μg/mL) antibodies for 24 h. For conjugation, Jurkat T cells were incubated with SEE-pulsed Raji B cells for 24 h. After incubation, supernatants were collected from activated T cells or conjugation and the amount of produced IL-2 was determined by ELISA DuoSet^®^ ELISA kit following manufacture’s instruction.

### 4.10. Western Blot

Incubated Jurkat cells (1 × 10^6^/mL) in the indicated conditions were harvested for lysis in RIPA buffer with phosphatase inhibitor for 30 min on ice. Lysates were centrifuged at 14,000 rpm for 30 min at 4 °C and approximately 30 μg of the lysate was separated on 8–12% SDS–PAGE gels. Proteins were transferred on PVDF membranes (Bio-Rad, Hercules, CA, USA). Membranes were blocked in 5% skim milk for 1 h, rinsed, and incubated with indicated primary antibodies in 3% skim milk overnight. Excess primary antibodies were discarded by washing the membrane five times with TBS-T and incubated with 0.1 μg/mL peroxidase-labeled secondary antibodies (against rabbit or mouse) for 2 h. After five washes in TBS-T, bands were visualized with ECL Western blotting detection reagents (Thermo Fisher Scientific, Waltham, MA, USA) with an ImageQuant LAS 4000 (GE healthcare, Chicago, IL, USA).

### 4.11. Measurement of Expressions of Surface Molecules by Flow Cytometry

The expressions of CD40L on T cells surface were determined by flow cytometry. Stimulated Jurkat T cells (5 × 10^5^/mL) with anti-CD3 and anti-CD28 antibodies for the indicated time were harvested and stained with anti-CD40L antibodies conjugated with APC for 30 min on ice. CD40L expressions on cells surface were acquired by flow cytometry and the expressions were shown in histogram graph. Mean fluorescence intensities were presented in line graph.

### 4.12. Conjugation Assay

CMFDA-stained Jurkat T cells (1 × 10^5^/mL) were pre-treated with 40 μM chrysophanol for 1 h. Simultaneously, CMRA-stained Raji B cells (1 × 10^5^/mL) were pulsed with 1 μg/mL SEE for 1 h. Then Jurkat T cells and Raji B cells were mixed and incubated for 30 min or 24 h. In some cases, anti-CD40L neutralizing antibodies were added to Jurkat T cells before incubation. Conjugations were acquired by flow cytometry in double-positive population. For microscopic imaging, cells were seeded in plate and incubated for 1 h or 24 h. Images were obtained by IncuCyte and three images were randomLy selected from each group. Conjugations between green cells (Jurkat T cells) and orange cells (Raji B cells) were counted from each selected images and percentage conjugated T cells conjugation was calculated with total T cells.

### 4.13. Statistics

Mean values ± SEM were calculated from the data acquired from three independent experiments performed on separate days and presented in graph. One-way ANOVA were used to obtain significance (*p* value). * indicates differences between indicated groups considered significant at *p* < 0.05.

## Figures and Tables

**Figure 1 ijms-21-06122-f001:**
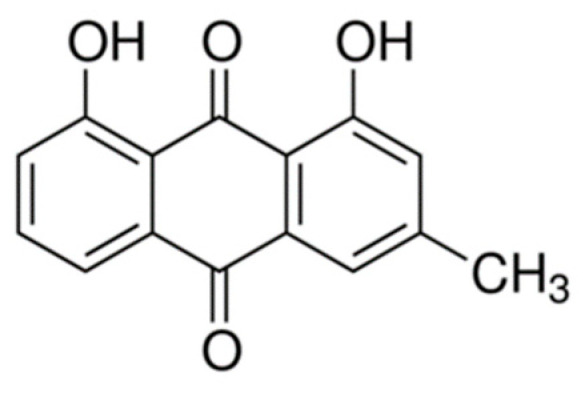
The chemical structure of chrysophanol.

**Figure 2 ijms-21-06122-f002:**
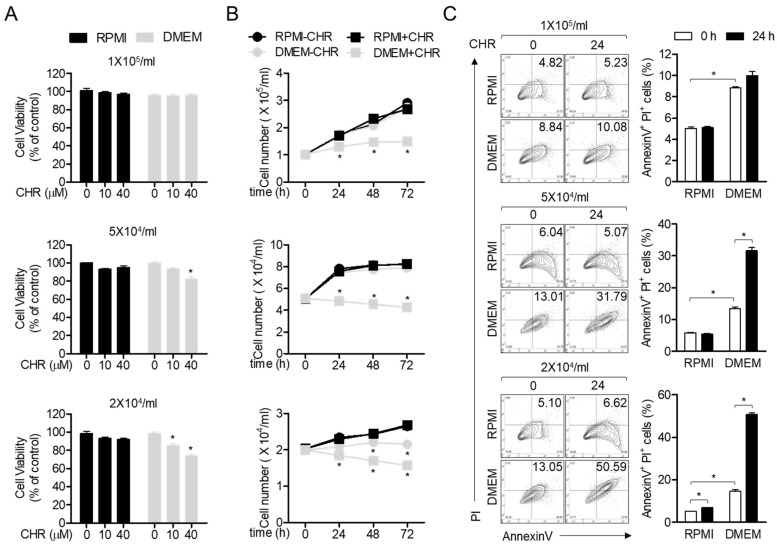
Chrysophanol is not cytotoxic to Jurkat T cells under culture condition using RPMI medium. (**A**) Cell viability of Jurkat T cells treated with the indicated concentrations of chrysophanol in the indicated media at the indicated density for 24 h was assessed by MTT (3-(4,5-dimethylthiazol-2-yl)-2,5-diphenyl tetrazolium bromide). (**B**) The growth rate of Jurkat T cells treated with/without 40 μM chrysophanol in the indicated media at the indicated density for the indicated time was measured by cell counting. (**C**) Apoptotic population of Jurkat T cells treated with 40 μM chrysophanol in the indicated media at the indicated density for 24 h was acquired by flow cytometry. The mean value of three experiments ± SEM (standard error of mean) is presented. * *p* < 0.05 versus control (**A**,**B**) or between two indicated groups (**C**). CHR—chrysophanol, RPMI—Rosewell Park Memorial Institute, DMEM—Dulbecco’s Modified Eagle Medium.

**Figure 3 ijms-21-06122-f003:**
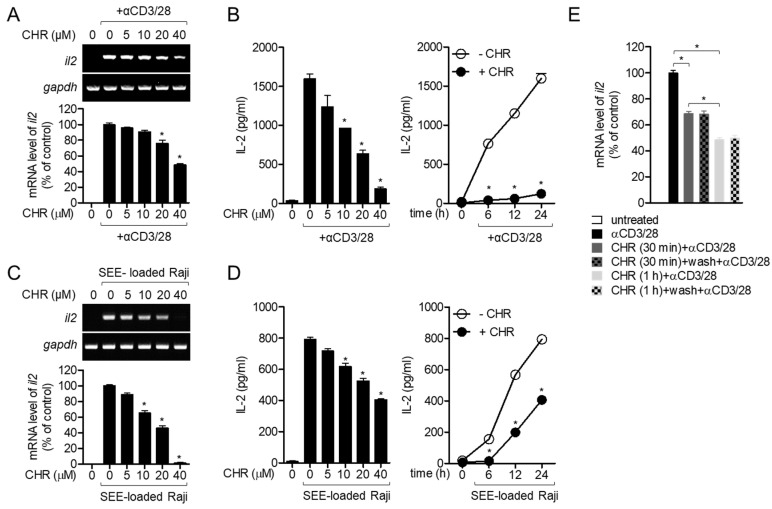
Chrysophanol inhibits IL-2 production from activated T cells. (**A**,**C**) Jurkat T cells were pre-treated with the indicated concentrations of chrysophanol for 1 h and stimulated with anti-CD3/CD28 antibodies (**A**) or SEE-loaded Raji B cells (**C**) for 6 h. Induced mRNA levels of *il2* gene were determined by conservative PCR (top) and quantitative PCR (bottom). (**B**,**D**) Jurkat T cells were pre-treated with the indicated concentrations (left) or 40 μM (right) of chrysophanol for 1 h and activated by anti-CD3/CD28 antibodies (**B**) or SEE-loaded Raji B cells (**D**) for 24 h (left) or 0–24 h (right). Produced IL-2 from activated Jurkat T cells was measured by ELISA. (**E**) Jurkat T cells were pre-treatment with 40 μM chrysophanol for 30 min or 1 h and washed with fresh media. Cells were stimulated with anti-CD3 and CD28 antibodies for 6 h then mRNA level of *il2* was assessed by quantitative PCR. The mean value of three experiments ± SEM is presented. * *p* < 0.05 between cells stimulated by anti-CD3/CD28 and cells indicated (**A**–**D**) and between two indicated groups (**E**). CHR—chrysophanol, SEE—staphylococcal enterotoxin E.

**Figure 4 ijms-21-06122-f004:**
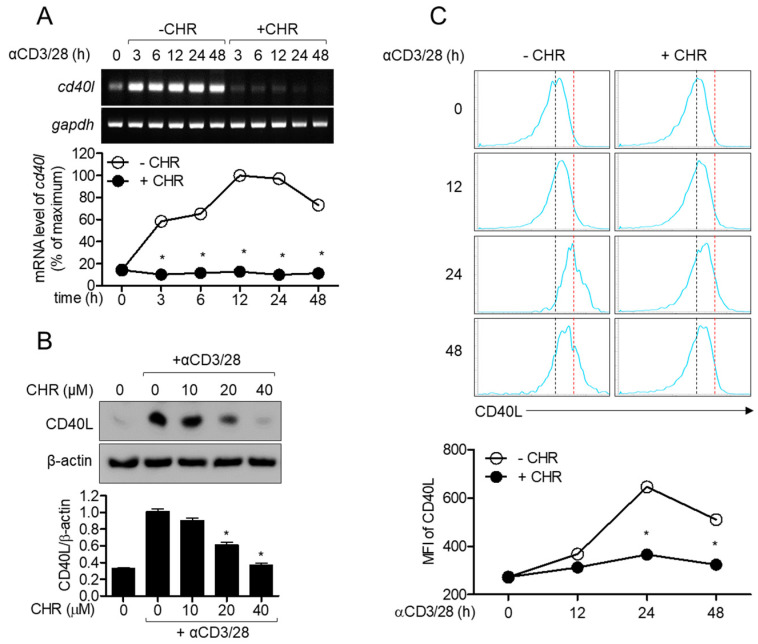
Chrysophanol suppresses the expression of CD40L in activated T cells. (**A**) Jurkat T cells were pre-treated with 40 μM of chrysophanol for 1 h and stimulated by anti-CD3/CD28 antibodies for 3 to 48 h. mRNA levels of *cd40l* were determined by conservative (top) and quantitative (bottom) PCR. (**B**) Jurkat T cells were pre-treated with the indicated concentrations of chrysophanol for 1 h and stimulated by anti-CD3/CD28 antibodies for 24 h. The expressed CD40Ls were detected by Western blot analysis, normalized to β-actin, and presented in a bar graph. (**C**) Jurkat T cells were pre-treated with 40 μM of chrysophanol for 1 h and stimulated by anti-CD3/CD28 antibodies for 12 to 48 h. The expressions of CD40L on cell surface were acquired by flow cytometry. Mean fluorescence intensities were presented in line graph. The mean value of three experiments ± SEM is presented. * *p* < 0.05 between cells stimulated by anti-CD3/CD28 and cells indicated. CHR—chrysophanol, MFI—Mean fluorescence intensity.

**Figure 5 ijms-21-06122-f005:**
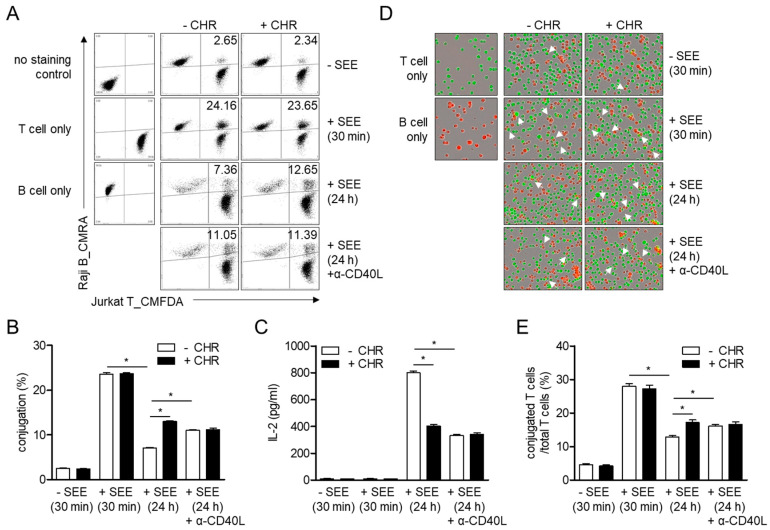
Abnormal expression of CD40L by chrysophanol leads to reduction of T cell activation through uncontrolled conjugation between B cells. (**A**–**E**) CMFDA (5-chloromethylfluorescein diacetate) stained Jurkat T cells were pre-treated with 40 μM of chrysophanol for 1 h and mixed with CMRA (5-(((4-chloromethyl)benzoyl)amino) tetramethylrhodamine) stained Raji B cells pulsed with SEE (staphylococcal enterotoxin E) for 30 min or 24 h. In some cases, anti-CD40L neutralizing antibodies (20 μg/mL) were added to Jurkat T cells. Conjugation was determined by flow cytometry (**A**) and double-positive population was presented in bar graph (**B**). Produced IL-2 from conjugates was assessed by ELISA (**C**). Cells were seeded in plates and microscopic images of conjugates between T cells and B cells were obtained by IncuCyte^®^ imaging system. White arrows indicate conjugates between T and B cells (**D**). The ratio of conjugated T cells was calculated in percentages of total T cells (**E**). The mean value of three experiments ± SEM (standard error of mean) is presented. * *p* < 0.05 between the two groups indicated. CHR—chrysophanol.

**Figure 6 ijms-21-06122-f006:**
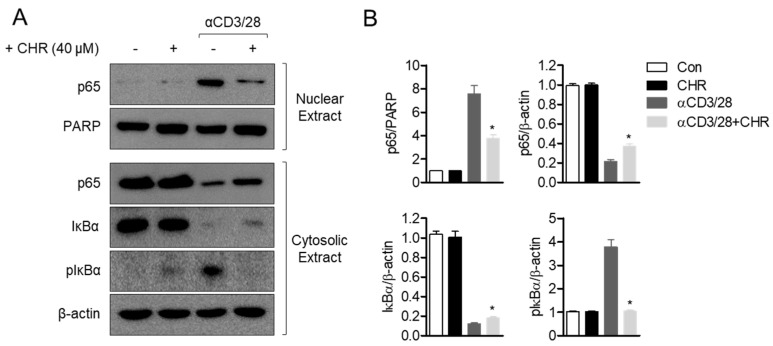
Treatment with chrysophanol blocks NFκB signaling pathway in activated T cells. (**A**) Jurkat T cells were pre-treated with 40 μM of chrysophanol for 1 h and stimulated by anti-CD3/CD28 antibodies for 1 h. Translocated p65, IκBα and phosphorylated IκBα were analyzed by Western blot from nuclear and cytosolic extracts. (**B**) Detected bands were normalized to PARP (poly (ADP-ribose) polymerase) for p65 in nuclear extract and β–actin for p65 in cytosolic extract, IκBα and phosphorylated IκBα. Normalization was shown in bar graph. The mean value of three experiments ± SEM is presented. * *p* < 0.05 between cells stimulated by anti-CD3/CD28 and cells indicated. CHR—chrysophanol, Con—control.

**Figure 7 ijms-21-06122-f007:**
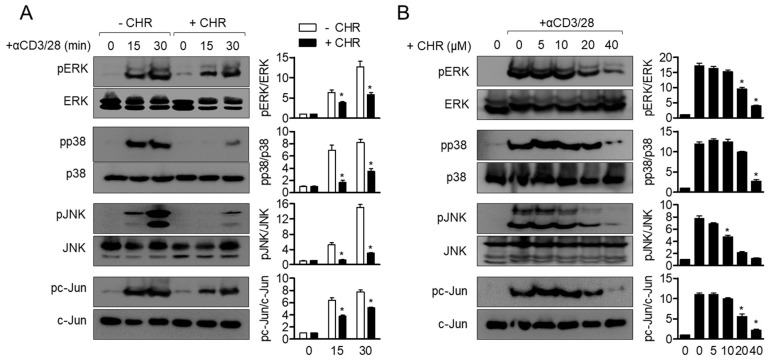
Pre-treatment with chrysophanol abrogates MAPK (mitogen activated protein kinase) signaling pathway in activated T cells. (**A**,**B**) Jurkat T cells were pre-treated with 40 μM of chrysophanol for 1 h and stimulated by anti-CD3/CD28 antibodies for 15 and 30 min (**A**). Jurkat T cells were pre-treated with the indicated concentrations of chrysophanol for 1 h and stimulated by anti-CD3/CD28 antibodies for 30 min (**B**). The phosphorylated levels of the indicated proteins were detected by Western blot analysis and normalized to the indicated total proteins. The mean value of three experiments ± SEM is presented. * *p* < 0.05 between cells stimulated by anti-CD3/CD28 and cells indicated. CHR—chrysophanol, ERK—extracellular signal-regulated kinases.

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
