# Peer review of "Chrysophanol Mitigates T Cell Activation by Regulating the Expression of CD40 Ligand in Activated T Cells"

_ijms, 2020, doi:10.3390/ijms21176122_

Round 1
Reviewer 1 Report
The manuscript entitled “Phytochemical chrysophanol mitigates T cell activation through the regulation of CD40L expression in activated T cells” investigate whether chrysophanol, a member of the anthraquinone family, regulates the activity of T cells. Authors demonstrated, in vitro, that chrysophanol exhert a suppressive effect on T cell activation through the regulation of CD40L expression. This original article could be a new contribution in the knowledge on the bioactivities of the chrysophanol.
Below are reported my comments the authors:
- The main doubt refers to the non-cytotoxic and non-apoptotic effect of chrysophanol described by the authors. Although few studies refer to the direct effect of this molecule on T cells, many articles describe the ability of chrysophanol to induce cell cycle arrest and apoptosis in tumor cells models. Furthermore, Yin J and colleagues, on 2019 demostrated that chrysophanol impacts on cell viability and apoptosis on Jurkat cells (Yin J, Yin Q, Liang B, et al. chrysophanol suppresses growth and metastasis of T cell acute lymphoblastic leukemia via miR-9 / PD-L1 axis. Naunyn Schmiedebergs Arch Pharmacol. 2020; 393 (2): 273-286. doi: 10.1007 / s00210-019-01778-0). The authors should discuss these discrepancy.
- Did the authors evaluate whether the chrysophanol-induced effect is reversible?
- Regarding the expression of cd40l and T-B cells conjugation, did the authors evaluate the effect induced by the T cells pre-treated with crhysophanol on the costimulatory molecules expressed bu APC?
- Authors should review the manuscript for grammar and syntax
Author Response
Response to reviewer’s comments
Reviewer #1.
Comments and Suggestions for Authors
The manuscript entitled “Phytochemical chrysophanol mitigates T cell activation through the regulation of CD40L expression in activated T cells” investigate whether chrysophanol, a member of the anthraquinone family, regulates the activity of T cells. Authors demonstrated, in vitro, that chrysophanol exhert a suppressive effect on T cell activation through the regulation of CD40L expression. This original article could be a new contribution in the knowledge on the bioactivities of the chrysophanol.
Below are reported my comments the authors:
- The main doubt refers to the non-cytotoxic and non-apoptotic effect of chrysophanol described by the authors. Although few studies refer to the direct effect of this molecule on T cells, many articles describe the ability of chrysophanol to induce cell cycle arrest and apoptosis in tumor cells models. Furthermore, Yin J and colleagues, on 2019 demostrated that chrysophanol impacts on cell viability and apoptosis on Jurkat cells (Yin J, Yin Q, Liang B, et al. chrysophanol suppresses growth and metastasis of T cell acute lymphoblastic leukemia via miR-9 / PD-L1 axis. Naunyn Schmiedebergs Arch Pharmacol. 2020; 393 (2): 273-286. doi: 10.1007 / s00210-019-01778-0). The authors should discuss these discrepancy.
Author’s response → We truly appreciate for raising important issue that cytotoxicity of chrysophanol on Jurkat T cells we used in the present study needs to be explained. We absolutely agree with your opinion that chrysophanol, a member of anthraquinone family, has been shown to possess anti-cancer activity, since it regulates proliferation and brings about apoptosis of cancerous cells. In particular, chrysophanol has been reported to cause cytotoxicity and pro-apoptotic activities in Jurkat T cells in the literature you provided [1]. By contrast, several studies have reported that chrysophanol does not exhibit cytotoxic effects and protect cells from critical damages [2–4]. It is obvious that cell viability or cytotoxicity assay including MTT and CCK-8 assay is highly dependent on the number of seeded cells and culture condition including media and FBS. To check this discrepancy whether chrysophanol has cytotoxicity on Jurkat T cells, we compared our culture conditions to the reference’s culture condition first. We found that Yin J and colleagues cultured Jurkat T cells in DMEM media, performed cytotoxicity assay with different density of cells and obtained Jurkat cells from different company. We cultured Jurkat T cells in RPMI media as ATCC’s instructions (Response fig. 1), performed experiments at the density of 1 × 104/200 μl/well. Under these condition, we evaluated that treatment with 40 μM chrysophanol for 24 h does not show cytotoxicity in Jurkat T cells and performed all experiments to demonstrate the effect of chrysophanol on T cell activity.
Response figure 1. Captured picture of ATCC website demonstrating culture method of Jurkat T cells. ATCC recommends that Jurkat T cells should be cultured in RPMI 1640 media with 10 % FBS if Jurkat T cells are maintained in healthy condition. A recommended cell density in culture is shown as 1 × 105 cells/ml.
To clarify whether treatment with chrysophanol exhibits cytotoxicity on Jurkat T cells cultured using different conditions as previously reported, we performed an MTT assay by comparing different media (RPMI versus DMEM) and different densities of cells (2 × 103/well to 1 × 104/well). Response figure 2A revealed that 40 μM chrysophanol did not exert cytotoxic effect on Jurkat T cells cultured in RPMI and DMEM at density of 1 × 104/well but displayed mild cytotoxicity to Jurkat T cells cultured only in DMEM at density of 5 × 103/well or 2 × 103/well. To obtain growth rate of Jurkat T cells in the presence of 40 μM chrysophanol, we counted the number of Jurkat T cells cultured in these two media every 24 h. As shown in Response figure 2B, Jurkat T cells cultured in DMEM showed a significant decrease in growth rate compared to Jurkat T cells cultured in RPMI. To confirm whether population of apoptotic cells induced by treatment with chrysophanol is dependent on culture media and cell number, AnnexinV/PI apoptosis assay was performed. Jurkat T cells cultured in DMEM exhibited increased apoptotic population compared to Jurkat T cells cultured in RPMI, however, treatment with chrysophanol has no pro-apoptotic at the density of 1 × 105/ml (response figure 2C, top panel). Interestingly, chrysophanol leads apoptotic pathway in DMEM cultured Jurkat T cells at the density of 5 × 104/ml and 2 × 104/ml (response figure 2C, middle and bottom panel). These result demonstrated that chrysophanol does not cause cell death and apoptosis in Jurkat T cells cultured in RPMI medium. We sincerely agree with your opinion that chrysophanol may be cytotoxic but we performed all experiments in the manuscript with condition which does not lead cytotoxicity in Jurkat T cells to explore the inhibitory effect of chrysophanol on T cell activity. We added these results in Figure 2 and revised ‘Abstract’, ‘Result’, ‘Discussion’, ‘Figure Legend’ and ‘Materials and Methods’ part. Please see the changes in the manuscript that highlight in red. Once again, we honestly appreciate your comment for giving us opportunity to consider very important point in our research.
Response figure 2. Chrysophanol is not cytotoxic to Jurkat T cells cultured in RPMI media. (A) Cell viability of Jurkat T cells treated with the indicated concentrations of chrysophanol in the indicated media at the indicated density for 24 h was assessed by MTT. (B) The growth rate of Jurkat T cells treated with 40 μM chrysophanol in the indicated media at the indicated density for the indicated time was measured by cell counting. (C) Apoptotic population of Jurkat T cells treated with 40 μM chrysophanol in the indicated media at the indicated density for 24 h was acquired by flow cytometry. The mean value of three experiments ± SEM is presented. *P < 0.05 versus control (A, B) or between two indicated groups (C).
- Did the authors evaluate whether the chrysophanol-induced effect is reversible?
Author’s response → We sincerely appreciate that you raised an important issue whether the effect of chrysophanol on T cell activation is reversible. To address your question whether the inhibited T cell activity by pre-treatment with 40 μM chrysophanol is reversed or recovered as the level of un-treated cells, we tested the removal effect of chrysophanol after pre-treatment with chrysophanol by comparing mRNA level of il2 from activated T cells pre-treated with chrysophanol for 30 min or 1 h only and activated T cells post-washed with fresh media after pre-treated with chrysophanol for 30 min or 1 h. As shown in response figure 3, pre-treatment with chrysophanol for 30 min and 1 h of T cells exerted significant decrease of mRNA level of il2 but removal of chrysphanol after pre-treatment did not show any attenuation in mRNA level of il2. Interestingly, decrease of mRNA level of il2 is dependent on the time of pre-treatment with chrysophanol, meaning that pre-treatment with chrysophanol for 1 h exhibited more inhibitory effect on T cell activation than pre-treatment for 30 min. This data demonstrated that pre-treatment with chrysophanol possesses a regulatory effect on T cell activation within 30 min and chrysophanol–induced effects are irreversible. This result is also consistent with previous report showing that chrysophanol has irreversible virucidal effect on poliovirus [5,6]. Hopefully this result would reach your expectation. We added this result in Figure 3E and revised ‘Result’ part. Please see the changes.
Response figure 3. Chrysophanol-induced inhibitory effect on T cell activation is irreversible. Jurkat T cells were pre-treatment with 40 μM chrysophanol for 30 min or 1 h and washed with fresh media. Cells were stimulated with anti-CD3 and CD28 antibodies for 6 h then mRNA level of il2 was assessed by quantitative PCR. The mean value of three experiments ± SEM is presented. *P < 0.05 between two indicated groups.
- Regarding the expression of cd40l and T-B cells conjugation, did the authors evaluate the effect induced by the T cells pre-treated with chrysophanol on the costimulatory molecules expressed by APC?
Author’s response → We honestly appreciate your valuable comments. As we discussed in the manuscript, T cells and B cells interact each other through T-B conjugation. CD40, CD80 and CD86 are expressed on B cell surfaces in resting condition and they play a critical role as costimulatory molecules by binding to CD40L and CD28 respectively on T cells. Since it has been reported that their expressions are induced dependent on B cell activation by T cells [7,8], we evaluated the mRNA levels of these co-stimulatory molecules on T-B conjugates whether decrement of T cell activity by pre-treatment with chrysophanol affects to the expression of co-stimulatory molecules on B cells. As shown in response figure 4, mRNA levels of cd40, cd80 and cd86 were inhibited in activated B cells by T cells pre-treated with chrysophanol. These results demonstrate that pre-treatment with chrysophanol of T cells has an influence on the expression of co-stimulatory molecules expressed on B cell surface and is also highly involved in B cell activity.
Response figure 4. Pre-treatment of Jurkat T cells with chrysophanol inhibits the expression of co-stimulatory molecules expressed on B cells in T-B conjugates. Jurkat T cells pre-treated with 40 μM chrysophanol were co-cultured with Raji B cells pulsed with 5 μg/ml SEE for 6 h. Then mRNA levels of cd40, cd80 and cd86 were measured by quantitative PCR analysis. The mean value of three experiments ± SEM is presented. *P < 0.05 versus control group.
- Authors should review the manuscript for grammar and syntax
Author’s response → To improve legibility and correct several grammar and syntax issues of our manuscript, we received English editing by professional editing company where native speakers revise scientific manuscripts. Please see the changes in the manuscript and certificates below.
References
- Yin, J.; Yin, Q.; Liang, B.; Mi, R.; Ai, H.; Chen, L.; Wei, X. Chrysophanol suppresses growth and metastasis of T cell acute lymphoblastic leukemia via miR-9/PD-L1 axis. Naunyn. Schmiedebergs. Arch. Pharmacol. 2020, 393, 273–286.
- Qian, Z.-J.; Zhang, C.; Li, Y.-X.; Je, J.-Y.; Kim, S.-K.; Jung, W.-K. Protective Effects of Emodin and Chrysophanol Isolated from Marine Fungus Aspergillus sp. on Ethanol-Induced Toxicity in HepG2/CYP2E1 Cells. 2011, 2011.
- Ueno, Y.; Umemori, K.; Niimi, E. ‐C; Tanuma, S. ‐I; Nagata, S.; Sugamata, M.; Ihara, T.; Sekijima, M.; Kawai, K. ‐I; Ueno, I.; et al. Induction of apoptosis by T‐2 toxin and other natural toxins in HL‐60 human promyelotic leukemia cells. Nat. Toxins 1995, 3, 129–137.
- Lee, M.S.; Sohn, C.B. Anti-diabetic properties of chrysophanol and its glucoside from rhubarb rhizome. Biol. Pharm. Bull. 2008, 31, 2154–2157.
- Semple, S.J.; Pyke, S.M.; Reynolds, G.D.; Flower, R.L.P. In vitro antiviral activity of the anthraquinone chrysophanic acid against poliovirus. Antiviral Res. 2001, 49, 169–178.
- Prateeksha; Yusuf, M.A.; Singh, B.N.; Sudheer, S.; Kharwar, R.N.; Siddiqui, S.; Abdel-Azeem, A.M.; Fraceto, L.F.; Dashora, K.; Gupta, V.K. Chrysophanol: A natural anthraquinone with multifaceted biotherapeutic potential. Biomolecules 2019, 9.
- Bromley, S.K.; Iaboni, A.; Davis, S.J.; Whitty, A.; Green, J.M.; Shaw, A.S.; Weiss, A.; Dustin, M.L. The immunological synapse and CD28-CD80 interactions. Nat. Immunol. 2001, 2, 1159–1166.
- Elgueta, R.; Benson, M.J.; De Vries, V.C.; Wasiuk, A.; Guo, Y.; Noelle, R.J. Molecular mechanism and function of CD40/CD40L engagement in the immune system. Immunol. Rev. 2009, 229, 152–172.
Reviewer 2 Report
The submitted manuscript from Lee & Jeong attempts to analyse the effects of chrysophanol on the viability and activation of a T cell line (Jurkats), using a range of approaches to investigate these questions. Broadly, there is some interesting data on the effects of chrysophanol, especially on the downstream signalling. However, I feel that some of the assays were performed in non-standard ways and I am unconvinced that chrysophanol is not affecting cell viability from their data. This has important consequences for explaining their subsequent signalling datasets.
Given that Jurkats are a suspension cell line, it was very surprising to see their viability being measured in terms of confluency… Cell counting is far more appropriate, although it would have been much more useful to have followed their growth rate in the presence of chrysophanol over several days to look for a significant growth defect. The AnnexinV and Caspase staining is not sufficiently sensitive using an IncuCyte system (there was rather an over-reliance on using this in general). Given that the authors had access to a flow cytometer, they should have performed these assays like almost everyone else in the field does!
A paper from the end of last year (https://doi.org/10.1007/s00210-019-01778-0) looked exactly at the effect of chrysophanol on Jurkat viability and function, using the same concentration range and timeframe as the authors did, and came to the opposite conclusion that chrysophanol was indeed detrimental to Jurkat viability. I would say their results are more convincing and if true, rather negate all the remaining data in the submitted manuscript since if the cells are apoptosing it would not be very surprising that signalling is downregulated. Those authors backed up their findings in a second T-ALL cell line as well, and also looked at downstream signalling too. The fact that this paper was not cited is troubling (found by a trivial google search of “chrysophanol T cell”).
Overall, I think my concerns about the cell viability data, along with the lack of novelty due to the paper already published means I am not very enthusiastic about this work. Significant effort would also need to be put in to improve the legibility of the text, as there are a number of places where the meaning of the authors is not entirely clear.
Author Response
Response to reviewer #2’s comments
Comments and Suggestions for Authors
The submitted manuscript from Lee & Jeong attempts to analyse the effects of chrysophanol on the viability and activation of a T cell line (Jurkats), using a range of approaches to investigate these questions. Broadly, there is some interesting data on the effects of chrysophanol, especially on the downstream signalling. However, I feel that some of the assays were performed in non-standard ways and I am unconvinced that chrysophanol is not affecting cell viability from their data. This has important consequences for explaining their subsequent signalling datasets.
- A paper from the end of last year (https://doi.org/10.1007/s00210-019-01778-0) looked exactly at the effect of chrysophanol on Jurkat viability and function, using the same concentration range and timeframe as the authors did, and came to the opposite conclusion that chrysophanol was indeed detrimental to Jurkat viability. I would say their results are more convincing and if true, rather negate all the remaining data in the submitted manuscript since if the cells are apoptosing it would not be very surprising that signalling is downregulated. Those authors backed up their findings in a second T-ALL cell line as well, and also looked at downstream signalling too. The fact that this paper was not cited is troubling (found by a trivial google search of “chrysophanol T cell”).
Author’s response → We truly appreciate for raising important issue that cytotoxicity of chrysophanol on Jurkat T cells we used in the present study needs to be explained. We absolutely agree with your opinion that chrysophanol, a member of anthraquinone family, has been shown to possess anti-cancer activity, since it regulates proliferation and brings about apoptosis of cancerous cells. In particular, chrysophanol has been reported to cause cytotoxicity and pro-apoptotic activities in Jurkat T cells in the literature you provided [1]. By contrast, several studies have reported that chrysophanol does not exhibit cytotoxic effects and protect cells from critical damages [2–4]. It is obvious that cell viability or cytotoxicity assay including MTT and CCK-8 assay is highly dependent on the number of seeded cells and culture condition including media and FBS. To check this discrepancy whether chrysophanol has cytotoxicity on Jurkat T cells, we compared our culture conditions to the reference’s culture condition first. We found that Yin J and colleagues cultured Jurkat T cells in DMEM media, performed cytotoxicity assay with different density of cells and obtained Jurkat cells from different company. We cultured Jurkat T cells in RPMI media as ATCC’s instructions (Response fig. 1), performed experiments at the density of 1 × 104/200 μl/well. Under these condition, we evaluated that treatment with 40 μM chrysophanol for 24 h does not show cytotoxicity in Jurkat T cells and performed all experiments to demonstrate the effect of chrysophanol on T cell activity.
Response figure 1. Captured picture of ATCC website demonstrating culture method of Jurkat T cells. ATCC recommends that Jurkat T cells should be cultured in RPMI 1640 media with 10 % FBS if Jurkat T cells are maintained in healthy condition. A recommended cell density in culture is shown as 1 × 105 cells/ml.
To clarify whether treatment with chrysophanol exhibits cytotoxicity on Jurkat T cells cultured using different conditions as previously reported, we performed an MTT assay by comparing different media (RPMI versus DMEM) and different densities of cells (2 × 103/well to 1 × 104/well). Response figure 2A revealed that 40 μM chrysophanol did not exert cytotoxic effect on Jurkat T cells cultured in RPMI and DMEM at density of 1 × 104/well but displayed mild cytotoxicity to Jurkat T cells cultured only in DMEM at density of 5 × 103/well or 2 × 103/well. To obtain growth rate of Jurkat T cells in the presence of 40 μM chrysophanol, we counted the number of Jurkat T cells cultured in these two media every 24 h. As shown in Response figure 2B, Jurkat T cells cultured in DMEM showed a significant decrease in growth rate compared to Jurkat T cells cultured in RPMI. To confirm whether population of apoptotic cells induced by treatment with chrysophanol is dependent on culture media and cell number, AnnexinV/PI apoptosis assay was performed. Jurkat T cells cultured in DMEM exhibited increased apoptotic population compared to Jurkat T cells cultured in RPMI, however, treatment with chrysophanol has no pro-apoptotic at the density of 1 × 105/ml (response figure 2C, top panel). Interestingly, chrysophanol leads apoptotic pathway in DMEM cultured Jurkat T cells at the density of 5 × 104/ml and 2 × 104/ml (response figure 2C, middle and bottom panel). These result demonstrated that chrysophanol does not cause cell death and apoptosis in Jurkat T cells cultured in RPMI medium. We sincerely agree with your opinion that chrysophanol may be cytotoxic and its cytotoxicity would make all signaling pathway downregulated in Jurkat T cells but we performed all experiments in the manuscript with condition which does not lead cytotoxicity in Jurkat T cells to explore the inhibitory effect of chrysophanol on T cell activity. We added these results in Figure 2 and revised ‘Abstract’, ‘Result’, ‘Discussion’, ‘Figure Legend’ and ‘Materials and Methods’ part. Please see the changes in the manuscript that highlight in red. Once again, we honestly appreciate your comment for giving us opportunity to consider very important point in our research.
Response figure 2. Chrysophanol is not cytotoxic to Jurkat T cells cultured in RPMI media. (A) Cell viability of Jurkat T cells treated with the indicated concentrations of chrysophanol in the indicated media at the indicated density for 24 h was assessed by MTT. (B) The growth rate of Jurkat T cells treated with 40 μM chrysophanol in the indicated media at the indicated density for the indicated time was measured by cell counting. (C) Apoptotic population of Jurkat T cells treated with 40 μM chrysophanol in the indicated media at the indicated density for 24 h was acquired by flow cytometry. The mean value of three experiments ± SEM is presented. *P < 0.05 versus control (A, B) or between two indicated groups (C).
- Given that Jurkats are a suspension cell line, it was very surprising to see their viability being measured in terms of confluency… Cell counting is far more appropriate, although it would have been much more useful to have followed their growth rate in the presence of chrysophanol over several days to look for a significant growth defect. The AnnexinV and Caspase staining is not sufficiently sensitive using an IncuCyte system (there was rather an over-reliance on using this in general). Given that the authors had access to a flow cytometer, they should have performed these assays like almost everyone else in the field does!
Author’s response → We sincerely appreciate your valuable comments. As shown in Author’s response #1, we replaced confluency data to growth rate by counting cell number every 24 h. Furthermore, AnnexinV/PI apoptosis assay was performed to replace IncuCyte data showing the expression of AnnexinV and caspase3/7. Please see the changes in Figure 2, ‘Result’ part, ‘Discussion’ and ‘Materials and Methods’ parts.
- Overall, I think my concerns about the cell viability data, along with the lack of novelty due to the paper already published means I am not very enthusiastic about this work. Significant effort would also need to be put in to improve the legibility of the text, as there are a number of places where the meaning of the authors is not entirely clear.
Author’s response → To improve legibility and correct several grammar and syntax issues of our manuscript, we received English editing by professional editing company where native speakers revise scientific manuscripts. Besides, we made revision in terms of discrepancy along with cytotoxicity of chrysophanol previously reported. Despite our manuscript did not reach your interests, we honestly appreciate that you reviewed and give us opportunities to revise our manuscript to improve it better. Hopefully our efforts to make our manuscript better in this revision time would meet your expectations. Please see the changes in the manuscript and certificate for English editing.
References
- Yin, J.; Yin, Q.; Liang, B.; Mi, R.; Ai, H.; Chen, L.; Wei, X. Chrysophanol suppresses growth and metastasis of T cell acute lymphoblastic leukemia via miR-9/PD-L1 axis. Naunyn. Schmiedebergs. Arch. Pharmacol. 2020, 393, 273–286.
- Qian, Z.-J.; Zhang, C.; Li, Y.-X.; Je, J.-Y.; Kim, S.-K.; Jung, W.-K. Protective Effects of Emodin and Chrysophanol Isolated from Marine Fungus Aspergillus sp. on Ethanol-Induced Toxicity in HepG2/CYP2E1 Cells. 2011, 2011.
- Ueno, Y.; Umemori, K.; Niimi, E. ‐C; Tanuma, S. ‐I; Nagata, S.; Sugamata, M.; Ihara, T.; Sekijima, M.; Kawai, K. ‐I; Ueno, I.; et al. Induction of apoptosis by T‐2 toxin and other natural toxins in HL‐60 human promyelotic leukemia cells. Nat. Toxins 1995, 3, 129–137.
- Lee, M.S.; Sohn, C.B. Anti-diabetic properties of chrysophanol and its glucoside from rhubarb rhizome. Biol. Pharm. Bull. 2008, 31, 2154–2157.
Round 2
Reviewer 1 Report
The authors answered the objections and resolved the critical issues. The new version of the manuscript is proposed in a comprehensive fashion, thus worthy of publication.
Author Response
Reviewer #1.
Comments and Suggestions for Authors
The authors answered the objections and resolved the critical issues. The new version of the manuscript is proposed in a comprehensive fashion, thus worthy of publication.
Author’s response → It was precious time to improve our manuscript according to your valuable comments. We sincerely appreciate your positive remarks about our manuscript.
Reviewer 2 Report
The authors have done a very good job at responding to the comments from both reviewers and credit must be given for this work. Also, the use of a proofreading company to polish the manuscript has made a very significant improvement to the readability of the work - money well spent...
I have a couple more points that will need to be addressed before accepting the manuscript that has arisen from the new data in figure 1. The authors use different cell densities to look at the cytotoxicity of CHR but in much of the results section and figure 2, this is defined as, say, 1x10^4 / well. What's a well unit? Nowhere in the methods does it say what plate they are using (assume 96 well) or the volume in the well so it is impossible to calculate the important value, which is the density in cells/ml. This is shown for Figure 2C though, confusingly. So the authors need to improve the methods and remove all references to cells/well and replace with cells/ml.
In a related point, the authors say they followed the ATCC guidelines for how to culture Jurkat cells. These instructions are very clear in that these cells should be maintained between 1x10^5 and 1x10^6 cells/ml. So why are most of the experiments done with cells at lower densities than the minimum recommended density - this makes no sense? The likely reason the cells die in DMEM at 2x10^4/ml is that they are too sparse and so not secreting enough autocrine growth factors. When the cells are grown at the recommended density (1x10^5/ml) there is no RPMI/DMEM difference in cell viability.
Finally, what was the growth rate of Jurkats in RPMI/DMEM in the absence of CHR? I agree that the difference between the two media in the presence of CHR is different. But does CHR decrease the growth rate when cells are in RPMI, when compared to the untreated cells? This is an important dataset that is missing from this figure and could easily have been added in.
Author Response
Response to reviewer #2’s comments
Comments and Suggestions for Authors
- The authors have done a very good job at responding to the comments from both reviewers and credit must be given for this work. Also, the use of a proofreading company to polish the manuscript has made a very significant improvement to the readability of the work - money well spent...
Author’s response → It was precious time to improve our manuscript according to your valuable comments. We sincerely appreciate your positive remarks about our new version of manuscript.
- I have a couple more points that will need to be addressed before accepting the manuscript that has arisen from the new data in figure 1. The authors use different cell densities to look at the cytotoxicity of CHR but in much of the results section and figure 2, this is defined as, say, 1x10^4 / well. What's a well unit? Nowhere in the methods does it say what plate they are using (assume 96 well) or the volume in the well so it is impossible to calculate the important value, which is the density in cells/ml. This is shown for Figure 2C though, confusingly. So the authors need to improve the methods and remove all references to cells/well and replace with cells/ml.
Author’s response → We truly appreciate for raising important issue that there is possibility to make readers confused by using two different units in the manuscript. To perform additional experiments for figure 2, we seeded cells at the density of 1 × 104/100 μl/96-well plate. To clarify unit issues, we improve ‘Materials and methods’ section and replace all ‘cell/well’ unit with ‘cell/ml’ unit (1 × 104/well → 1 × 105/ml) as you suggested. Please see the changes in the manuscript.
- In a related point, the authors say they followed the ATCC guidelines for how to culture Jurkat cells. These instructions are very clear in that these cells should be maintained between 1x10^5 and 1x10^6 cells/ml. So why are most of the experiments done with cells at lower densities than the minimum recommended density - this makes no sense? The likely reason the cells die in DMEM at 2x10^4/ml is that they are too sparse and so not secreting enough autocrine growth factors. When the cells are grown at the recommended density (1x10^5/ml) there is no RPMI/DMEM difference in cell viability.
Author’s response → We absolutely agree with your comment that our descriptions make confusion. We maintained Jurkat T cells with density of 5 × 105/ml to 1 × 106/ml using RPMI medium by following ATCC’s instruction and used cells for experiments within this range (cell viability assay: 1 × 105/ml, PCR experiment: 5 × 105/ml, ELISA assay: 1 × 105/ml, Western blot assay: 1 × 106/ml, flow cytometry assay: 5 × 105/ml, conjugation assay: 1 × 105/ml). Please see the updated changes in manuscript showing how many cells are used for each experiment.
- Finally, what was the growth rate of Jurkats in RPMI/DMEM in the absence of CHR? I agree that the difference between the two media in the presence of CHR is different. But does CHR decrease the growth rate when cells are in RPMI, when compared to the untreated cells? This is an important dataset that is missing from this figure and could easily have been added in.
Author’s response → We appreciate your valuable comment that growth rate of Jurkat T cells in different media was missing. We added the result of growth rate of Jurkat T cells in RPMI/DMEM media to figure 2B as below. Please see the changes in manuscript.
Round 3
Reviewer 2 Report
The authors have now satisfied my final concerns about the paper